# High anisotropy in electrical and thermal conductivity through the design of aerogel-like superlattice (NaOH)$_{0.5}$NbSe$_2$

Ruijin Sun [1] ✉, Jun Deng [2], Xiaowei Wu[2], Munan Hao[2], Ke Ma[2,3], Yuxin Ma[2], Changchun Zhao[1], Dezhong Meng[1], Xiaoyu Ji[2,4], Yiyang Ding[5], Yu Pang[6], Xin Qian[6], Ronggui Yang[6], Guodong Li[2], Zhilin Li[2], Linjie Dai [7], Tianping Ying [2], Huaizhou zhao [2], Shixuan Du [2], Gang Li [2], Shifeng Jin[2,3] ✉ & Xiaolong Chen [2,3,8] ✉

Interlayer decoupling plays an essential role in realizing unprecedented properties in atomically thin materials, but it remains relatively unexplored in the bulk. It is unclear how to realize a large crystal that behaves as its monolayer counterpart by artificial manipulation. Here, we construct a superlattice consisting of alternating layers of NbSe$_2$ and highly porous hydroxide, as a proof of principle for realizing interlayer decoupling in bulk materials. In (NaOH)$_{0.5}$NbSe$_2$, the electric decoupling is manifested by an ideal 1D insulating state along the interlayer direction. Vibration decoupling is demonstrated through the absence of interlayer models in the Raman spectrum, dominant local modes in heat capacity, low interlayer coupling energy and out-of-plane thermal conductivity (0.28 W/mK at RT) that are reduced to a few percent of NbSe$_2$'s. Consequently, a drastic enhancement of CDW transition temperature (>110 K) and Pauling-breaking 2D superconductivity is observed, suggesting that the bulk crystal behaves similarly to an exfoliated NbSe$_2$ monolayer. Our findings provide a route to achieve intrinsic 2D properties on a large-scale without exfoliation.

Since the discovery of graphene[1], atomic-thick materials have attracted tremendous interest in the past decades[2–5]. Compared to the bulk, isolated monolayers show a rich variety of distinct properties owing to their interlayer decoupling, both electrically and vibrationally. For instance, graphene displays exceptional electronic transport properties and stiffness[6]. Monolayer NbSe$_2$ host exotic electronic states, including drastically enhanced charge-density wave (CDW) order and Ising superconductivity[7,8]. Black phosphorus[9] and MoS$_2$[2] undergo indirect to direct band gap transitions as their thickness is reduced to the monolayer limit. These examples demonstrate that interlayer decoupling significantly impacts mechanical, electrical, and optical properties. However, when considering aspects such as fabrication, chemical and environmental stability, and sample quantity, bulk materials continue to outperform their monolayer counterparts. Therefore, it is highly desired to find a route to realize interlayer decoupling in large crystals, so that intrinsic 2D properties can be realized in bulk materials.

At present, feasible routes have been developed to realize interlayer decoupling in monolayer or few-layer 2D materials[10–14]. However, this ability is notably absent in bulk materials. For an exfoliated

[1]School of Science, China University of Geosciences, Beijing (CUGB), 100083 Beijing, China. [2]Institute of Physics, Chinese Academy of Science, 100190 Beijing, China. [3]School of Physical Sciences, University of Chinese Academy of Sciences, 100190 Beijing, China. [4]School of Physics, Liaoning University, 110136 Shenyang, China. [5]Department of Physics, Imperial College London, London SW7 2AZ, UK. [6]School of Energy and Power Engineering, Huazhong University of Science and Technology, 430074 Wuhan, China. [7]Cavendish Laboratory, 19 JJ Thomson Avenue, Cambridge CB3 0HE, UK. [8]Songshan Lake Materials Laboratory, 523808 Dongguan, China. ✉e-mail: srj@cugb.edu.cn; shifengjin@aphy.iphy.ac.cn; chenx29@iphy.ac.cn

monolayer, the vibrational and electrical connections to the bulk are interrupted by the surrounding vacuum or air (Fig. 1a). In the case of stacked bilayer or thicker few atomic-layer materials, the interlayer coupling can be partially removed by increasing the twist angle[15,16] or introducing lattice mismatch[17] (Fig. 1b). However, in conventional bulk crystals, their periodic symmetry does not accommodate variations in either twist angle or lattice mismatch. Although artificially decoupling a bulk crystal remains a challenge, computational chemistry regularly obtains properties of decoupled 2D monolayers by introducing thick (~2 nm) vacuum layers into the periodic lattice of bulk materials. This vacuum layer, which inhibits the interlayer transmission of both electrons and phonons, serves as an ideal block layer for decoupling the bulk crystal into isolated monolayers. In practical applications, if the vacuum layer could be replaced by materials that are equally insulating to electrons and phonons, perfect decoupling of the bulk material should be achievable. Recently, this concept was partially demonstrated in a bulk superlattice, $Ba_6Nb_{11}S_{28}$, wherein the metallic $NbS_2$ layers were spatially separated by 1 nm thick $Ba_3NbS_5$ block layers, resulting in clean 2D superconductivity[18]. However, akin to other metals inserted with insulating block layers, metallic behavior persists along the interlayer direction of $Ba_6Nb_{11}S_{28}$, indicating imperfect electron decoupling. To the best of our knowledge, there is currently no evidence that traditional block layers can prevent phonon transmissions and decouple the bulk material vibrationally.

In practice, few solid materials share the same capability of blocking phonons as the vacuum. For example, it is well known that the thermal conductance of a crystalline material, which is mediated by non-localized phonons, cannot be arbitrarily low[19]. Surprisingly, for some highly-porous-and-low-density materials, such as aerogels, their thermal conductivity (12 mW m$^{-1}$ K$^{-1}$) can be even lower than that of air[20–22]. Inspired by the extraordinarily hollow and disordered structure of aerogels, we construct some hydroxide layers with extremely high porosity, i.e., up to 95% of the volume in some hydroxide layers remains vacant. These fluffy aerogel-like layers (~2 nm) facilitate the lattice matching with a variety of transition metal dichalcogenides (TMDs) and give rise to a new family of bulk superlattices (Fig. S1). In particular, $(NaOH)_{0.5}NbSe_2$, one of the superlattices with alternating porous NaOH block layers and $NbSe_2$ monolayers, is selected as a model system to investigate the effect of artificial interlayer decoupling in bulk materials (Fig. 1c).

$NbSe_2$ is one of the most studied van der Waals materials, and meanwhile, a model system for investigating the effect of

dimensionality on correlated electron-phonon phenomena. In bulk $NbSe_2$, a CDW sets in at $T_{CDW} = 33$ K, and superconductivity sets in at a critical temperature $T_C = 7.2$ K[23]. In the monolayer limit, however, a highly unusual enhancement of $T_{CDW}$ is observed, and a Pauli-breaking Ising superconductivity set in at 1.9 K[7,8,24]. In this work, we show that those exotic monolayer behaviors are indeed realized in large single crystals of $(NaOH)_{0.5}NbSe_2$. Systematic investigations suggest that the thick and porous NaOH layers act as superinsulators and detach the $NbSe_2$ monolayers spatially, electrically and vibrationally, as confirmed by X-ray diffraction, Raman spectrum, heat capacity, heat transport measurements, orientation and temperature-dependent resistivity, and density functional theory (DFT) calculations. Moreover, the fitting of angle-dependent magnetic resistance and magnetic torque data reveal that the exotic superconductivity and CDW states in bulk $(NaOH)_{0.5}NbSe_2$ are consistent with those found in $NbSe_2$ monolayer, suggesting this decoupled bulk crystal act just like an exfoliated $NbSe_2$ monolayer. The concept of introducing aerogel-like layers to decouple the bulk materials can be easily extended to other TMDs materials (Fig. S1) and even beyond (graphite, Xenes, BN, etc.), which may bring us to a new era of unconventional bulk materials with unprecedented electrical, mechanical and optical properties.

## Results and discussion
### Chemical compositions and crystal structure
As shown in Figs. 2a and S1, single crystals of $(AOH)_xMX_2$ (A = Na, K; M = Ta, Nb; X = S, Se) up to 0.6 cm in size were synthesized under hydrothermal conditions by using $MX_2$ single crystals as the precursor (detailed in methods). In the case of $(NaOH)_{0.5}NbSe_2$, the X-ray diffraction data in Fig. 2a show that the (00l) reflections of the $NbSe_2$ precursors are completely replaced by a new set of ones that systematically shift to lower angles, suggesting the incorporation of new block layers into the $NbSe_2$ layered structure. Figure S2 shows the scanning electron microscopy (SEM) image for the as-prepared crystal, EDS mapping reveals a homogeneous distribution of Na, Nb, and Se atoms after the reaction. The atomic ratio of Na: Nb: Se is determined to be 0.5:1:2, so the concentration x of Na-containing species should be half of the $NbSe_2$. XPS analysis was carried out on a freshly cleaved single crystal to probe the species and the oxidation states of the elements in $NbSe_2$ and $(NaOH)_{0.5}NbSe_2$, with the corresponding Nb$3d$, Se$3d$, Na$1s$, and OH$^-$ spectra presented in Fig. 2b. The Nb$3d$ and Se$3d$ spectra of pristine and the intercalated $NbSe_2$ have identical binding energies, indicating the oxidization state of $NbSe_2$ is unaltered upon

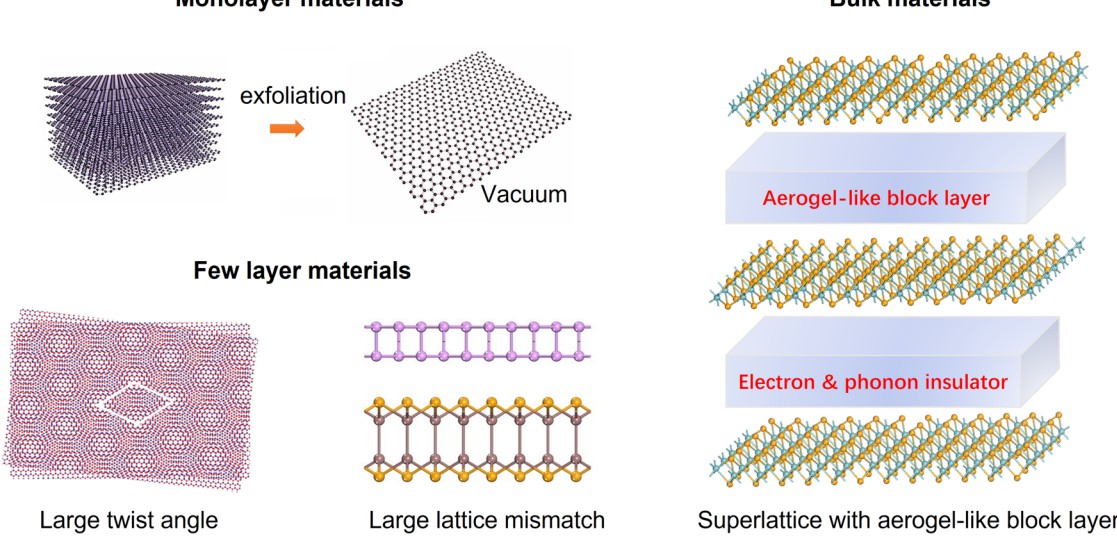

**Fig. 1 | Interlayer decoupling schemes for different materials.** The design scheme for achieving interlayer decoupling through exfoliating monolayers, introducing large twist angle and lattice mismatch in a few atomic layers, and incorporating aerogel-like block layers in bulk materials.

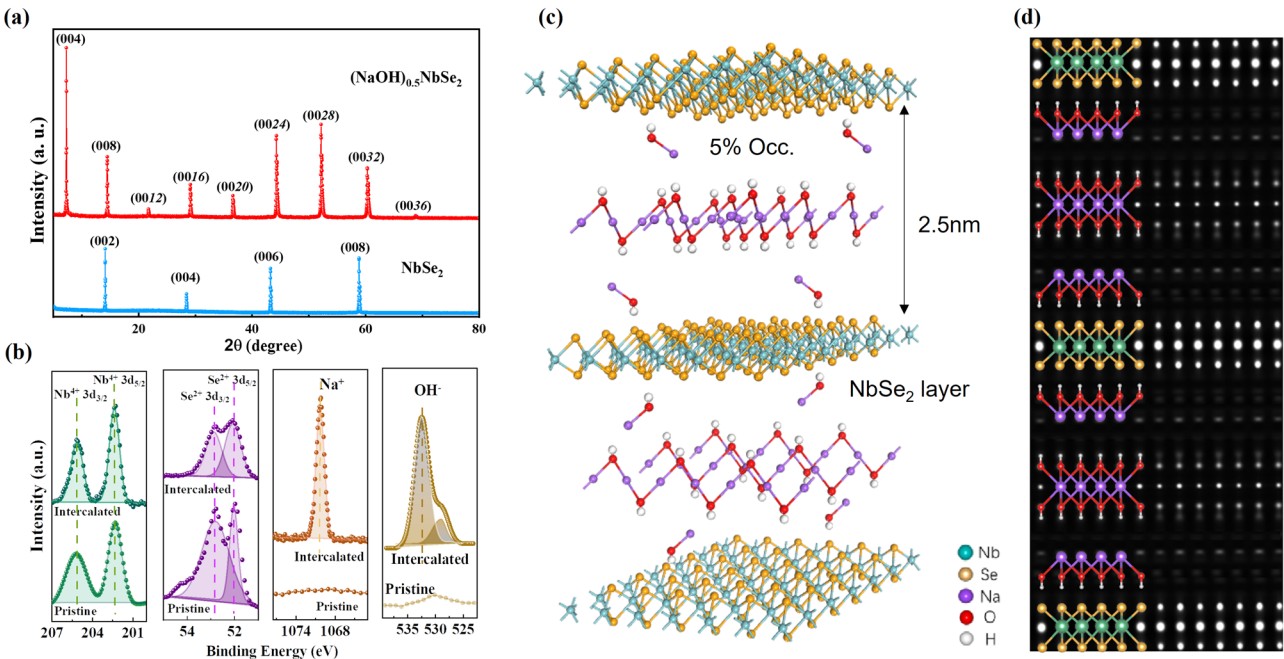

**Fig. 2 | Structure characterization of (NaOH)$_{0.5}$NbSe$_2$. a** Powder X-ray diffraction patterns collected for NbSe$_2$ and (NaOH)$_{0.5}$NbSe$_2$ single crystals. **b** XPS pattern of Nb, Se, and Na, OH$^-$ for (NaOH)$_{0.5}$NbSe$_2$. **c** Crystal structure of (NaOH)$_{0.5}$NbSe$_2$. **d** Experimental charge density of (NaOH)$_{0.5}$NbSe.

reaction, and there is no evidence of charge transfer between the block layer and the NbSe$_2$ matrix. Meanwhile, a sharp peak at 1070 eV was ascribed to the Na$^+$[25], and the peak at 532.4 eV was assigned to OH$^-$, confirming the NaOH molecules as the intercalants. The shoulder peak at 529.1 eV corresponded to a trace amount of adsorbed water molecules, rather than hydrated water in the structure[26]. From this, the chemical formula of the intercalated samples can be best described as (NaOH)$_{0.5}$NbSe$_2$.

To determine the crystal structures of (NaOH)$_{0.5}$NbSe$_2$, single-crystal X-ray diffraction (SXRD) analysis was implemented on a small specimen (132 × 74 × 17 µm³). The SXRD diffraction data can be indexed by a trigonal unit cell (space group P-3m1), with lattice parameters $a = 3.4554$ Å, $c = 48.9648$ Å (Fig. 2c). Form the high accuracy experimental electron density constructed via. the model independent Maximum entropy method (Fig. 2d), the NaOH molecules can be located at the 1a (0, 0, 0), 1b (0, 0, 1/2), 2c (0, 0, z), and 2d (2/3, 1/3, z) sites, which form ~2 nm thick NaOH trilayers in between the NbSe$_2$ monolayers. It is worth noting that the resolved electron density of the two outmost NaOH layers is much lower (Fig. 2d), suggesting all the sites neighboring the NbSe$_2$ layers have much smaller occupancy factors. Based on the resolved structure model, the crystal structure refinements converged rapidly to small residuals $R1 = 0.0313$. The final crystal structures and the refined structural parameters are shown in Fig. 2c & d and Table S1, and the bond lengths and angles in (NaOH)$_{0.5}$NbSe$_2$ are close to that of 2H-NbSe$_2$ and NaOH (Table S2). As shown in Fig. 2c & d, the extraordinarily large NbSe$_2$ interlayer distance ($d = 24.5$Å) is up to four times that of pristine NbSe$_2$ (6.14 Å), and two times that of most known intercalated NbX$_2$ (X = S, Se) compounds[27], confirming the NbSe$_2$ monolayers are well spatially detached. More strikingly, structure refinement suggests only 5 % of the sites in the two outmost layers are randomly occupied by NaOH molecules, consistent with the results of charge density analysis (Fig. 2c, d). The density of the extremely low occupied two NaOH layers surrounding the NbSe$_2$ layer is merely 0.08 g cm$^{-3}$, comfortably staying in the typical density range of aerogels (C.A. 0.0011 to ~0.5 g cm$^{-3}$). The occupancy of the central NaOH layer reaches 40 %, which is necessary to brace up the structure framework.

## The uniaxial-insulating behavior and electrical decoupling

Electrical anisotropy is generally small in metallic crystals. However, in the case of a fully decoupled metal, an insulating behavior is expected along the interlayer direction, whereas metallic behavior should be retained for in-plane directions. The MX$_2$(M = Ta, Nb, X = S, Se) single crystals used here are all metallic[28], and the reported electrical anisotropy $\beta = \rho_{ab}(T)/\rho_c(T)$ is lower than 200 ($\rho_{ab}(T)$ and $\rho_c(T)$ are in-plane and out-plane resistivities, respectively)[29]. By inserting ~ 2 nm thick AOH layers in between the MX$_2$ layers, giant electronic anisotropy appeared in all (AOH)$_x$MX$_2$ materials, which behave as either band insulators or metals, depending on the measured crystal axis (Figs. S3 and S4a). Figures 3a, S4b show the $\rho_{ab}(T)$ and $\rho_c(T)$ data of NbSe$_2$ and (NaOH)$_{0.5}$NbSe$_2$ measured between 3 K and 300 K. Along both a- and c-directions, the NbSe$_2$ single crystal show metallic behavior, where the measured β was negligibly small ~ less than 2.5 at 300 K. At 7 K, a superconducting transition is observed (Fig. S4b). After incorporating the thick NaOH block layers, the (NaOH)$_{0.5}$NbSe$_2$ crystal remains metallic along the a-direction, and the resistivity increases slightly compared to that of NbSe$_2$. Along the c-axis, however, the resistivity increases drastically—up to 10$^6$ times that of NbSe$_2$. Furthermore, the rapid enhancement of $\rho_c(T)$ at low temperature can be well fitted by a semiconducting thermal activation model (Fig. S3). Interestingly, this exotic axial-dependent insulating behavior is observed in all the measured (AOH)$_x$MX$_2$ materials, as shown in Fig. S3. In Fig. S5, the resistance anisotropy β of various TMDs materials are presented. The β of the metallic TMDs (NbSe$_2$, TaS$_2$, etc.) is very small, generally in the order of ~10, while their β can be further increased to ~10$^3$–10$^4$ by the intercalation of molecules or inorganic layers[30,31]. The recent incorporation of 1 nm thick, stoichiometric Ba$_3$NbS$_5$ layers in quasi-2D superlattice Ba$_6$Nb$_{11}$S$_{28}$[18] increased the β values to around 10$^3$, which is three orders of magnitudes smaller than the maximum value realized in (NaOH)$_{0.5}$NbSe$_2$. The huge resistance anisotropy and especially the insulating behavior of $\rho_c(T)$ suggest that the thick hydrate layers have effectively eliminated the interlayer electronic couplings between NbSe$_2$ monolayers.

Figure 3b shows the DFT calculated band structures of (NaOH)$_{0.5}$NbSe$_2$, bulk NbSe$_2$ and monolayer NbSe$_2$. For (NaOH)$_{0.5}$NbSe$_2$,

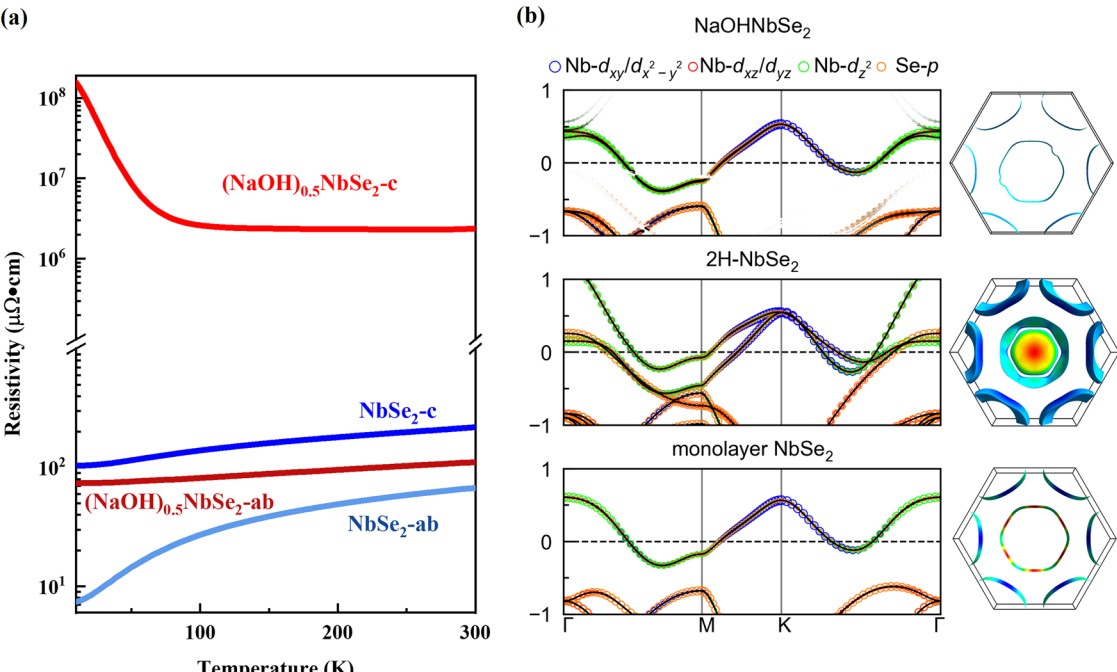

**Fig. 3 | The uniaxial-insulating behavior and band structure of (NaOH)$_{0.5}$NbSe$_2$.** **a** The temperature-dependent in-plane and out-plane resistivity measured for NbSe$_2$ and (NaOH)$_{0.5}$NbSe$_2$ between 3 and 300 K. **b** The band structures and Fermi surfaces of (NaOH)$_{0.5}$NbSe$_2$, bulk, and monolayer NbSe$_2$. The size of the circles is proportional to the contribution of NbSe$_2$.

all bands crossing the Fermi level are contributed by the spatially detached NbSe$_2$ layers, resulting in nearly identical band structures and fermi surfaces of (NaOH)$_{0.5}$NbSe$_2$ and monolayer NbSe$_2$. In bulk NbSe$_2$, the degeneracy of the Nb $3d$ band that crosses the Fermi level is broken by the interlayer coupling with bonding/antibonding configurations of Se $p_z$ orbitals, which gives rise to two split bands in the Γ-M-K-Γ plane and two concentric cylindrical Fermi surface. In the monolayer form, the interlayer coupling is absent, both bands shifting of Se $p_z$ orbitals across the Fermi level and the band splitting of Nb $4d$ orbitals are avoided, leaving only one Nb band across the Fermi level. In (NaOH)$_{0.5}$NbSe$_2$, the Se $p_z$ conducting bands are found to sink below the Fermi energy, and the band degeneracy of Nb $4d$ states is almost perfectly preserved, leading to a monolayer-like band structure without interlayer couplings[32]. Moreover, in bulk NbSe$_2$, the antibonding between interlayer Se $p_z$ orbitals lift the Se $3p$ band across the Fermi level, creating a small Fermi surface with strong $k_z$ dispersion. In (NaOH)$_{0.5}$NbSe$_2$, the band structure exhibits only the dispersionless flat bands along the Γ−A direction. Therefore, like monolayer NbSe$_2$, the fermi surfaces of (NaOH)$_{0.5}$NbSe$_2$ are derived from a single (or degenerate) Nb $4d$ band that is dispersionless along the $k_z$ direction, confirming the elimination of interlayer electronic couplings.

### The vibrational and phonon decoupling

Besides electronic decoupling, we demonstrate that the numerous NbSe$_2$ layers in (NaOH)$_{0.5}$NbSe$_2$ separated by sparse NaOH layers are also decoupled vibrationally. Figure 4a shows the room-temperature Raman spectra for bulk NbSe$_2$, monolayer NbSe$_2$[7] and (NaOH)$_{0.5}$NbSe$_2$, respectively. The NbSe$_2$ intra-layer vibration models, including E$_{2g}$ mode at ~250 cm$^{-1}$, A$_{1g}$ mode at ~220 cm$^{-1}$, and the soft mode at the range from 150 to 200 cm$^{-1}$, were observed in all three samples. However, the most known interlayer vibration model in bulk and few-layer NbSe$_2$, which manifest itself as a strong low-frequency vibrational band (20–30 cm$^{-1}$), is absent in monolayer NbSe$_2$ and (NaOH)$_{0.5}$NbSe$_2$. The absence of low-frequency shearing modes in the Raman spectrum of NbSe$_2$ is only found in its monolayer form, implying that in (NaOH)$_{0.5}$NbSe$_2$, interlayer vibration couplings between NbSe$_2$ monolayers are effectively eliminated by introducing thick aerogel-like NaOH layers.

The vibrationally decoupled substructures in (NaOH)$_{0.5}$NbSe$_2$ can be treated as independent harmonic oscillators (or Einstein oscillators), similar to clathrates and filled-skutterudites, whereas the remaining lattice can be treated within the Debye model. Figure 4b shows the temperature-dependent heat capacity $C_p$ of (NaOH)$_{0.5}$NbSe$_2$ in the range of 2–300 K, with a fitting curve using a combined Debye–Einstein mode (detailed in supplementary materials). A minimum of two Einstein modes are required to adequately model the temperature dependence of $C_p$, whose characteristic temperatures were determined to be $\theta_{E1} = 128K$ and $\theta_{E2} = 174K$ through fitting, with the estimated Debye temperature $\theta_{Debye} = 237K$ (see the fitting parameters for Debye–Einstein model in Table S3). As shown in Fig. 4b, the harmonic Einstein oscillators contribute to 91% of the total specific heat, suggesting a substantial part of phonon vibrations in (NaOH)$_{0.5}$NbSe$_2$ is localized. Considering the chemical bonds within the NbSe$_2$ layers are intact and strong, we believe the localized phonon is confined by the weak interlayer bonding between the sparse NaOH layers and NbSe$_2$ layers, with the absence of up to 95% of the chemical bonds.

To experimentally verify the vibrational decoupling in (NaOH)$_{0.5}$NbSe$_2$ along the interlayer direction, the thermal conductivity of a (NaOH)$_{0.5}$NbSe$_2$ single crystal is measured using the beam-offset time domain thermos-reflectance (TDTR) method[33]. In Fig. 4c, we compare the ratio of the in-phase voltage ($V_{in}$) to the out-of-phase voltage ($V_{out}$) of the measured TDTR signals with the ratio calculated from a thermal model. As shown in Fig. 4c, at room temperature the out-of-plane thermal conductivity κ$_\perp$ is determined to be merely 0.28 W m$^{-1}$ K$^{-1}$ within experimental uncertainties, which is drastically reduced to 7% of bulk NbSe$_2$ (Fig. S6). The through-plane conductivity of (NaOH)$_{0.5}$NbSe$_2$ is among the lowest values achieved in bulk inorganic materials[34–37]. The sharp decline of κ$_\perp$ in (NaOH)$_{0.5}$NbSe$_2$ can be attributed to the ultralow interlayer coupling strength of (NaOH)$_{0.5}$NbSe$_2$, which leads to the much-reduced group velocity of lattice vibrations along the out-of-plane direction. In order

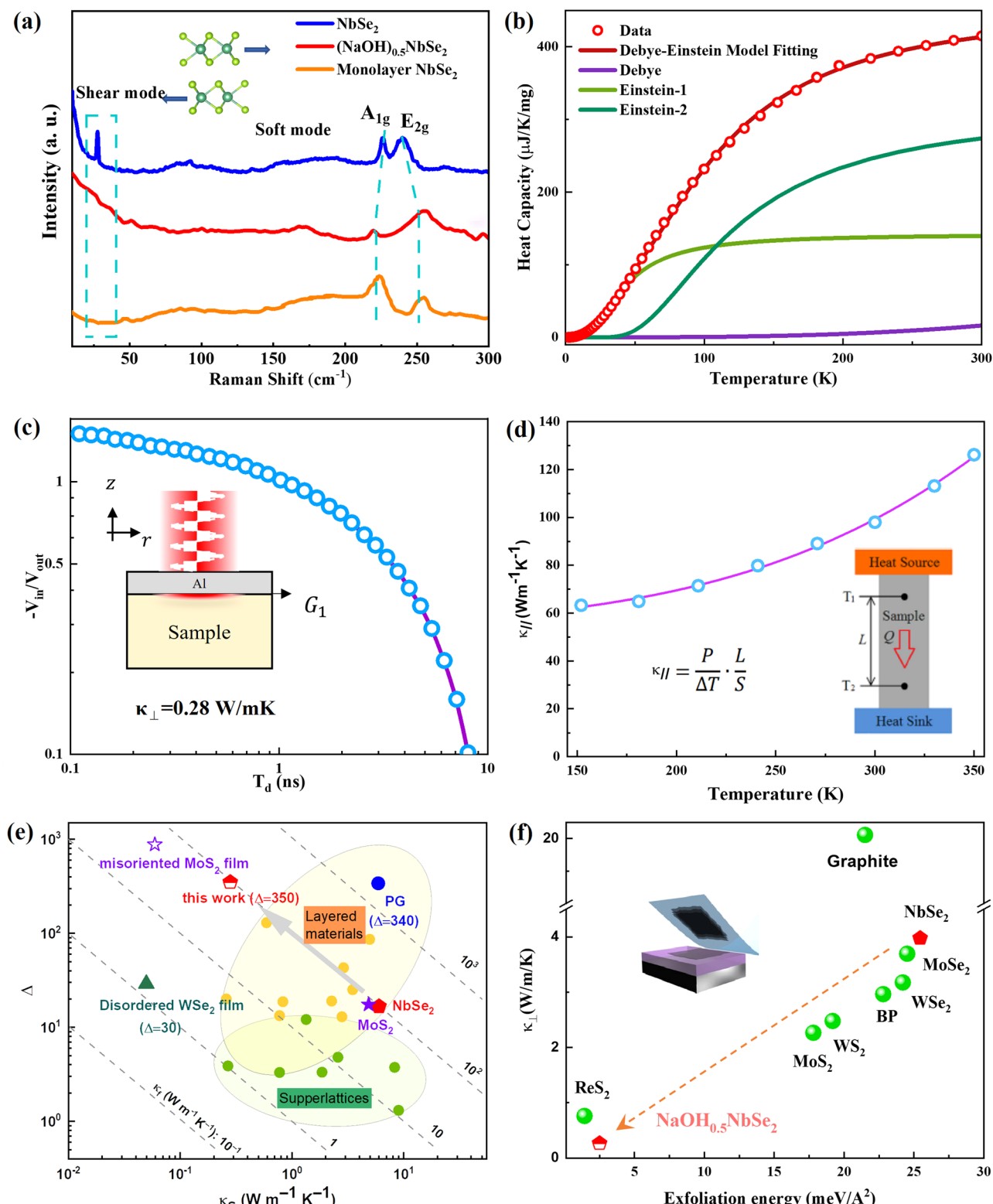

**Fig. 4 | The vibrational and phonon decoupling behaviors of $(NaOH)_{0.5}NbSe_2$.**
**a** Room-temperature Raman spectra for bulk, monolayered $NbSe_2$ and
$(NaOH)_{0.5}NbSe_2$. **b** $C_p$ vs. temperature plot in 2–300 K range. The solid blue line is
calculated using combined Debye–Einstein model. The individual contributions
from Debye ($\beta$) and the two Einstein terms are also plotted. **c** TDTR data for the out-
of-plane thermal conductivity $\kappa_\perp$ of $(NaOH)_{0.5}NbSe_2$. **d** the in-plane thermal
conductivity $\kappa_{//}$ of $(NaOH)_{0.5}NbSe_2$ measured by steady-state heat flow method.
**e** Comparison of $\Delta = \kappa_{//}/\kappa_\perp$, $\kappa_{//}$ (x axis), and $\kappa_\perp$ (diagonal dashed lines) measured
for highly anisotropic thermal conductors and $(NaOH)_{0.5}NbSe_2$. Data of layered
materials and superlattice are taken from Ref. 38. **f** $\kappa_\perp$ vs. exfoliation energy dia-
gram for typical TMDs materials, graphite, black phosphorus (BP) and
$(NaOH)_{0.5}NbSe_2$[33,38–40,49–51].

to evaluate the anisotropy of thermal conductivity, we measured the in-plane thermal conductivity $\kappa_{\parallel}$ of $(NaOH)_{0.5}NbSe_2$ single crystal using a steady-state heat flow setup (Fig. 4d). The $\kappa_{\parallel}$ of $(NaOH)_{0.5}NbSe_2$ was measured in the temperature range of 150–350 K, and the results are presented in Fig. 4d. At 300 K, the measured $\kappa_{\parallel}$ is 98.1 W/mK, suggesting the thermal conductivity anisotropy ($\Delta = \kappa_{//}/\kappa_{\perp}$) of $(NaOH)_{0.5}NbSe_2$ reaches an exceptionally high value of ~350. As shown in Fig. 4e, this large anisotropic value is second only to $MoS_2$ thin films stacked by misoriented monolayers ($\Delta \approx 880$)[38], and exceeds that of pyrolytic graphite (PG)—one of the most anisotropic bulk thermal conductors ($\Delta \approx 340$)[39].

To get an insight into the observed interlayer decoupling at the atomic level, we have calculated the potential energy curves by elongating the interlayer distance between $NbSe_2$ atomic layers, which gives the theoretical interlayer coupling energy E. As shown in Fig. S7a, the DFT calculations show that $NbSe_2$ monolayers are indeed weakly coupled to NaOH layers in $(NaOH)_{0.5}NbSe_2$. The interlayer coupling energy E only reached 2.5 meV/ $Å^2$, around 9 % of that of bulk $NbSe_2$ (25 meV/$Å^2$). Such a small E is even comparable to the low coupling energy that found in exfoliated monolayers deposited on different substrates[40]. Naturally, the low occupancy factor in aerogel-like NaOH layers (Occ = 0.05) is expected to reduce the strength of the vdWs interactions. To clarify how partial occupancy affects the interlayer coupling energy, we performed DFT calculations based on structure models with a range of Occ between 0.05 and 1.0. As shown in Fig. S7b, the interlayer energy E increased proportionally to Occ, and reached 29 meV/$Å^2$ at Occ = 1.0, a value similar to that of pristine $NbSe_2$. The findings suggest the ultralow density (or Occ) in aerogel-like NaOH layers plays a crucial role in reducing the interlayer coupling in $(NaOH)_{0.5}NbSe_2$. Figure 4f presents a survey of crystallized vdWs materials sorted according to their interlayer coupling energy and $\kappa_{\perp}$, both of which should approach 0 when the material is perfectly decoupled. It shows that $(NaOH)_{0.5}NbSe_2$ exhibits both very low interlayer coupling energy and $\kappa_{\perp}$, and more than 90% of them in pristine $NbSe_2$ is removed by incorporating the highly porous NaOH layers, pushing $(NaOH)_{0.5}NbSe_2$ toward the best bulk material in terms of interlayer decoupling.

**The enhanced CDW and suppressed superconductivity**

The CDW transition usually involves lattice distortion in which the electron-phonon coupling plays an important role. In bulk $NbSe_2$, the CDW transition is generally below 33 K[7]. In monolayer $NbSe_2$, the electron-phonon interactions are significantly enhanced in the 2D limit, leading to a remarkable enhancement of the charge density wave (CDW) phenomenon ($T_{CDW} = 145K$)[7]. In line with this, Raman scattering experiments have revealed the emergence of two new phonon vibrational modes in the CDW state compared to the normal state. This is evident in the spectrum by the presence of two additional vibrational peaks (at 190 $cm^{-1}$ and 75 $cm^{-1}$) under $T_{CDW}$. In Figs. 5a and S8, we show the temperature dependence of the Raman spectra measured for $(NaOH)_{0.5}NbSe_2$. From 300 K to 140 K, the spectra are relatively clean and agree well with monolayer $NbSe_2$ above its $T_{CDW}$. The prominent features below 300 $cm^{-1}$ include an $E_2g$ mode around 250 $cm^{-1}$, an $A_1g$ mode around 220 $cm^{-1}$, and a soft model around 170 $cm^{-1}$, with the absence of the interlayer shear model in bulk $NbSe_2$. Below 140 K, two new modes appear in the Raman spectra. The first new mode emerged around 190 $cm^{-1}$, and the second feature is a broad amplitude mode that appeared around 75 $cm^{-1}$, both of which were observed in the CDW-state of bulk $NbSe_2$ and monolayer $NbSe_2$ below $T_{CDW}$ and were attributed to the collective excitation of the CDW fluctuations. As shown in Fig. 5a, in $(NaOH)_{0.5}NbSe_2$, the CDW-related vibrational modes appeared much higher than 33 K, indicating an enhanced $T_{CDW}$. Figure 5b shows the intensity $I_{CDW}$ of the CDW-related model at 190 $cm^{-1}$. The $T_{CDW}$ of $(NaOH)_{0.5}NbSe_2$ should be below the temperature where $I_{CDW}$ dropped to zero. The temperature dependence of $I_{CDW}$

in bulk and monolayer $NbSe_2$ taken from pieces of literature are also presented in Fig. 5b for comparison[7]. As for $(NaOH)_{0.5}NbSe_2$, the intensity of the CDW-related Raman peak vanishes between 110 and 140 K, and the significantly enhanced CDW order is in line with the findings in monolayer $NbSe_2$. Moreover, the amplitude mode and folding mode corresponding to CDW in $(NaOH)_{0.5}NbSe_2$ are much stronger than in monolayer $NbSe_2$[7]. Since the intensity of the Raman signal is proportional to the number of layers, the enhancement here is likely to be caused by the superposition of many monolayer signals.

With the advantages of bulk crystals, we can directly use torque magnetometry to reveal more details about the CDW in $(NaOH)_{0.5}NbSe_2$. Torque magnetometry measures the torque $\tau = \mu_0 V M \times H$ that a magnetic moment **M** experienced in a uniform magnetic field **H** ($\mu_O$ is the permeability of vacuum), where the $\tau$ value is proportional to sample volumes V. This technique is particularly useful for studying CDW systems because the CDW order often leads to changes in the magnetic susceptibility. In the case of H-$NbSe_2$, the CDW order is known to arise within the Nb atomic layer. Both H-$NbSe_2$ and $(NaOH)_{0.5}NbSe_2$ feature Nb atomic layers with a D3h symmetry, which can be equally described in an orthogonal lattice using new crystallographic axes, namely $a_o$ and $b_o$ (Fig. S9a). In the orthorhombic setting, the torque $\tau$ is simplified to a periodic function of the doubled azimuthal angle 2φ measured from the a-axis:

$$\tau_{2\varphi} = \frac{1}{2}\mu_0 H^2 V \left[ (\chi_{aa} - \chi_{bb}) \sin 2\varphi - 2\chi_{ab} \cos 2\varphi \right] \qquad (1)$$

where the $\chi_{aa}$, $\chi_{bb}$ and $\chi_{ab}$ are magnetic susceptibility tensors in the orthorhombic setting. We performed torque measurements on 2H-$NbSe_2$ and $(NaOH)_{0.5}NbSe_2$ single crystals using a setup depicted in Fig. S9b. The φ-scan torque values were recorded between 10 K and 200 K, and the corresponding angular-dependent torque curves are presented in Fig. S10. Notably, the angular-dependent torque curves of $NbSe_2$ maintained a two-fold Sine function (with a period of π) throughout the CDW transition. This observation is consistent with the retention of D3h symmetry in the 3 × 3 CDW phase of $NbSe_2$. Below the CDW transition temperature, we observed a significant increase in the amplitude of $\tau_{2\varphi}$, indicating enhanced magnetic anisotropy between different susceptibility tensors in the CDW phase (Fig. 5c). For $(NaOH)_{0.5}NbSe_2$, the φ-scan magnetic torque spectra in Fig. 5d show the same $sin(2\varphi)$ form in both the normal state and below the $T_{CDW}$ (as determined by Raman scattering), strongly suggesting that the CDW phase in $(NaOH)_{0.5}NbSe_2$ also retained the D3h symmetry as in the 3×3 CDW phase of H-$NbSe_2$. Meanwhile, the drastic increase of magnetic torque strength, signifying the onset of CDW ordering, occurred at much higher temperatures in $(NaOH)_{0.5}NbSe_2$ (>120 K), in consistency with the Raman scattering results.

At last, we turn to the superconducting state of $(NaOH)_{0.5}NbSe_2$. Figure S11a shows a detailed view of the superconducting transition, which shows onset near $T = 1.7K$ and reaches zero resistance at $T = 1.2K$. The reduction in the transition temperature for the $NbSe_2$ layers in $(NaOH)_{0.5}NbSe_2$ relative to bulk H-$NbSe_2$ is consistent with $NbSe_2$ monolayer ($T_c = 1.5$ K)[24]. Noticeably, the $\rho_{ab}(T)$ for $(NaOH)_{0.5}NbSe_2$ for 1.5 K < T < 1.8 K is well described by the Halperin-Nelson model[41]: $\rho_{ab}^F(T) = \rho_{ab}^N(T) \exp(-b\sqrt{t})$, where $\rho_{ab}^F(T)$ and $\rho_{ab}^N(T)$ are the fluctuations and normal-state resistivity, respectively; $t = \left(\frac{T}{T_{HN}}\right) - 1$; and b is a fitting parameter on the order of 1. The Halperin-Nelson behavior proves fluctuations of the superconducting order parameter above a 2D Berezinskii–Kosterlitz–Thouless (BKT) transition.

Despite the decrease in $T_c$, the exotic Pauli-breaking Ising superconductivity appeared in monolayer $NbSe_2$, few-layer $NbSe_2$ and $Ba_6Nb_{11}S_{28}$ vdWs superlattice has attracted intensive attentions[8]. For a superconductor, applying an external magnetic field can suppress its superconductivity either via. the Zeeman effect or through in-plane

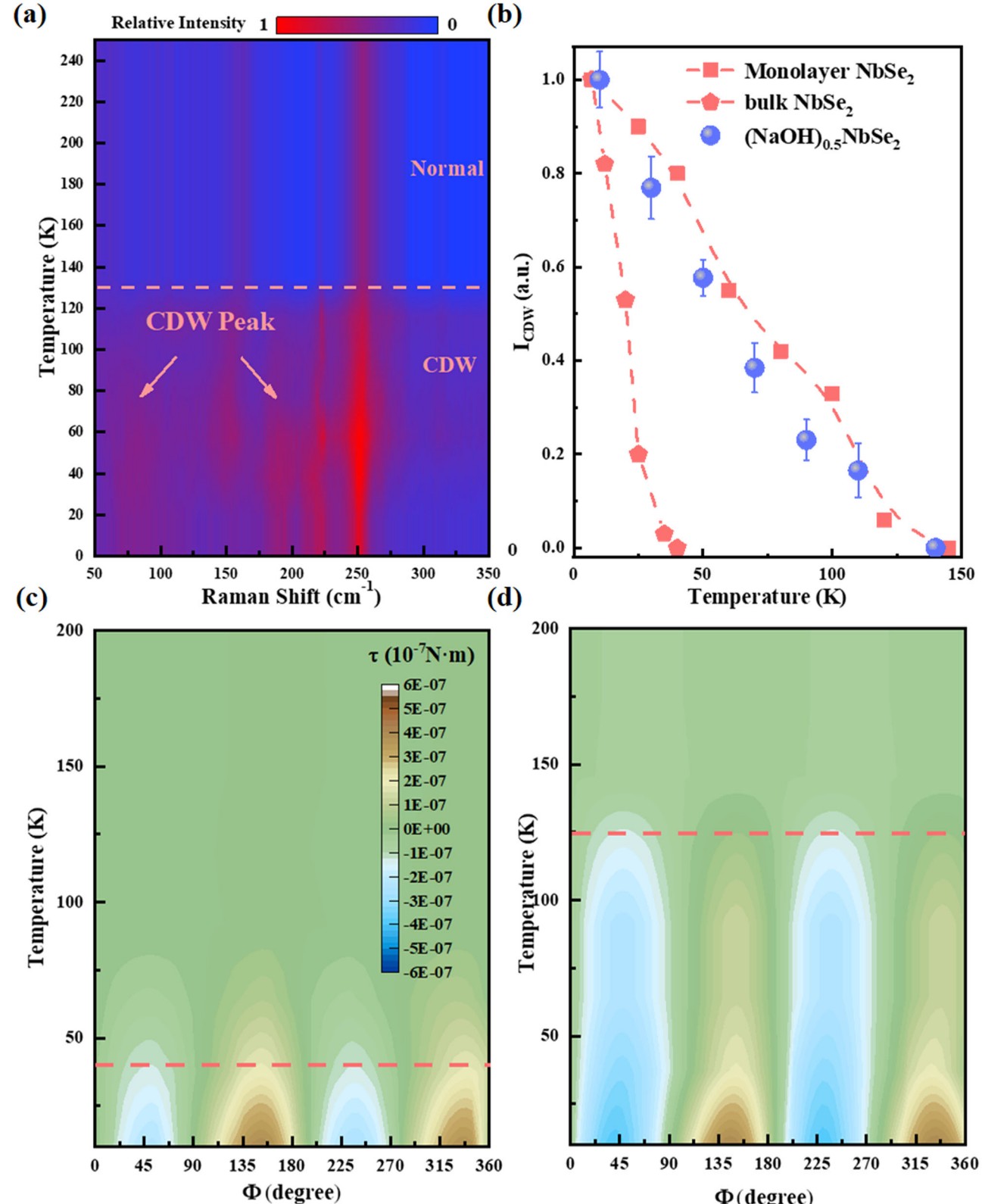

**Fig. 5 | CDW characterization of (NaOH)$_{0.5}$NbSe$_2$ and NbSe$_2$. a** The temperature-dependent Raman spectra for (NaOH)$_{0.5}$NbSe$_2$. **b** Temperature dependence of the amplitude mode intensity $I_{CDW}$ for (NaOH)$_{0.5}$NbSe$_2$ and reported monolayer NbSe$_2$ results. **c, d** The angular-dependent magnetic torque under various temperatures for bulk (**c**) NbSe$_2$ and (**d**) (NaOH)$_{0.5}$NbSe$_2$. The color mapping in the figure represents the torque value.

orbital effects. The Zeeman effect, in particular, imposes an upper bound on the critical magnetic field, known as the Pauli (or Clogston−Chandrasekhar) limit ($\mu_0H_p = 1.84T_c^{zero}$)[42]. The Pauli violation ratio (PVR), defined as $\mu_0H_{c\parallel}(O)/\mu_0H_P$, is generally smaller than 1.0

for conventional superconductors, including bulk NbSe$_2$. In monolayer NbSe$_2$, Pauli-limit violation Ising-superconductivity can be achieved in the presence of finite-momentum pairing or strong spin−orbit coupling (SOC)[8]. In bulk superlattice Ba$_6$Nb$_{11}$S$_{28}$ and few-layer NbSe$_2$,

some exotic Pauli-breaking Fulde–Ferrell–Larkin–Ovchinnikov (FFLO) state and orbital-FFLO state 2D superconductivity are also reported recently[18,43]. In $(NaOH)_{0.5}NbSe_2$, we show that its 2D superconducting behavior is more like the Ising-superconductivity appeared in monolayer $NbSe_2$. In Fig. S11d, the $\mu_0 H_{c2}$-$T/T_c$ diagram for $(NaOH)_{0.5}NbSe_2$ clearly shows that the Pauli violation ratio $\mu_0 H_{c2}^{ab}(0)/\mu_0 H_p$ reached a respectable high value up to 4.0, close to that of monolayer $NbSe_2$ and well above the PVR values in $Ba_6Nb_{11}S_{28}$ and few-layer $NbSe_2$[18,43]. In terms of anisotropy in both the normal state resistivity (β) and superconductivity (γ), the values in $(NaOH)_{0.5}NbSe_2$ ($\beta_{max} > 2,000,000$, $\gamma = 32.5$) are much higher than that in the 'layer-in-cake' 2D superconductors $Ba_6Nb_{11}S_{28}$ ($\beta_{max} \approx 2,000$, $\gamma = 26.34$) and $Ba_6Nb_{11}Se_{28}$ ($\beta_{max} \approx 20.0$, $\gamma = 10.3$)[18,31]. Moreover, considering the absence of sharp shapes of $\mu_0 H_{c2}^{ab}(0)$ data observed in FFLO systems of few-layer $NbSe_2$ and $Ba_6Nb_{11}S_{28}$, and the high value of PVR that also exceeds the limit of Rashba-type SOC, it is reasonable to assign the Pauli-breaking superconductivity in $(NaOH)_{0.5}NbSe_2$ to the effect of Ising paring without FFLO, a mechanism that well acknowledged in monolayer $NbSe_2$. Above all, the drastic enhancement of CDW above 110 K as well as the occurrence of exotic Pauli-breaking 2D superconductivity below 1.5 K, consistently suggests that the $(NaOH)_{0.5}NbSe_2$ bulk crystal behave just like a decoupled $NbSe_2$ monolayer.

In conclusion, we have developed a method to fabricate aerogel-like superlattices by incorporating hydrate molecules AOH (A = Na, K) into $MX_2$ (M = Ta, Nb; X = S, Se), resulting in large single crystals consisting of ~2 nm thick, porous hydrate layers and $MX_2$ monolayers. The elimination of interlayer electron couplings turns the metallic parent materials into 'insulators' along the interlayer direction, whereas along other crystallographic axes the metallic state remains. The vanished interlayer electron couplings are also supported by the ideal 2D electronic band structure and fermi surfaces. Fascinatingly, the rarely occupied (5%) hydrate layers also cut off the interlayer vibrational coupling in bulk $(NaOH)_{0.5}NbSe_2$, leading to an extremely low interlayer coupling energy of 2.5 meV/Å² and interlayer thermal conductivity of 0.28 W m⁻¹ K⁻¹. The disappearance of the interlayer shearing vibrational model, the appearance of two CDW-related models well above 110 K and the exotic Pauli-breaking 2D superconductivity below 1.5 K are nearly identical to the behaviors found in $NbSe_2$ monolayers. Our findings suggest the insertion of highly porous (aerogel-like) block layers can effectively decouple the monolayers in 3D crystal lattices, so that intrinsic 2D properties can be realized in conventional bulk materials.

## Methods
### Sample synthesis
For the first step, $MX_2$ (M = Ta, Nb; X = S, Se) crystals were grown from M powder of 99.95 % purity and X pellets of 99.999 % purity by iodine 99.8 % vapor transport in a gradient of 730–700 °C in a sealed quartz tube for 15 days. Then, 0.3–0.6 g of thiourea (alfa, 99.9% purity) and 1–2 g AOH (A = Na, K) (alfa, 99. 9% purity) were dissolved in 10 ml of ammonium sulfide aqueous solution (14 % in water) in a Teflon-lined stainless-steel autoclave (volume 25 mL). Then, several pieces of $MX_2$ crystals were added to the solution. Finally, the autoclave was sealed and heated up to 100–130 °C for 130 h. Upon recovery, large $(NaOH)_{0.5}NbSe_2$ single crystals with a different silver metallic luster were obtained by leaching and clearing.

### Structural characterization and composition determination
Powder X-ray diffraction (PXRD) patterns were collected at room temperature on a Rigaku smart Lab X-ray diffractometer operated at 40 kV voltage and 40 mA current using Cu Ka radiation (λ = 1.5406 Å). The 2θ range was 10–80°with a step size of 0.01. Indexing and Rietveld refinement were performed using the DICVOL91, Fullprof, and MDI Jade programs. Single crystal X-ray diffraction (SCXRD) patterns at

295 K were collected using a Bruker D8 VENTURE PHOTO II diffractometer with multilayer mirror monochromatized Mo Kα (λ = 0.71073 Å) radiation. Unit cell refinement and data merging were performed using the SAINT program, and an absorption correction was applied using multi-Scans scanning. Structural solutions with the P -3 m 1 space groups were obtained by intrinsic phasing methods using the program APEX3, and the final refinement was completed with the Jana 2020 suite of programs. The electron density maps of the two samples were first constructed by the charge flipping method implemented in the Jana2020 software. Scanning electron microscopy (SEM) images were taken on a Phenom pro XL microscope equipped with an electron microprobe analyzer for the semiquantitative elemental analysis in the energy-dispersive X-ray spectroscopy (EDS) mode.

### Magnetic torque measurements
The torque measurements used commercial components provided by Quantum Design Company (TRO Torque Magnetometer). As illustrated in the inset of Fig. S9, the single crystal is mounted onto a piezo-resistive cantilever before measuring. The applied magnetic field is 1 T.

### The density functional theory (DFT) calculations
were performed within the Vienna ab initio simulation package[44]. We adopted the generalized gradient approximation (GGA) in the form of Perdew-Burke-Ernzerhof (PBE) for the exchange-correlation potential[45]. The projector augmented-wave (PAW) pseudopotentials were used with a plane wave energy of 500 eV; 4s24p4 for Se, 4p64d45s1 for Nb, 2p63s1 for Na, 2p42s2 for O and 1s1 for H electron configuration were treated as valence electrons. A Monkhorst-Pack[46] Brillouin zone sampling grid with a resolution of 0.02 × 2πÅ⁻¹ in the self-consistent calculation and 0.01 × 2πÅ⁻¹ for Fermi surface calculation were applied. The position of H in $(NaOH)_{0.5}NbSe_2$ was manually added and the position of H and O were relaxed until all the forces on them were less than 0.1 eV/Å. The lattice constants were derived from the experimental results. The van der Waals interaction was considered using the DFT-D3 method of Grimme[47,48]. The total energy of the system as a function of interlayer separation is calculated by using GGA-PBE (Perdew–Burke–Ernzerhof) functional, as implemented in the CASTEP code. The kinetic energy cutoff for the plane-wave basis set was 410 eV. In the self-consistent potential and total energy calculations of NbSe2 layers, a set of (8 × 8 × 1) k-point samplings was used for Brillouin zone integration. The convergence criterion of self-consistent calculations for ionic relaxations was 10⁻⁴ eV between two consecutive steps. By using the conjugate gradient method, all atomic positions and unit cells were optimized until the atomic forces were 0.05 eV/Å.

### Thermal conductivity measurements
The out-of-plane thermal conductivity was measured using time-domain thermoreflectance (34). The pump and probe beam optical beams are focused on the surface of the samples using a microscope objective lens. The thermal conductivity is determined by comparing the time dependence of the ratio of the in-phase $V_{in}$ and the out-of-phase $V_{out}$ signals from the rf lock-in amplifier to calculations using a thermal model. The in-plane thermal conductivity on a $(NaOH)_{0.5}NbSe_2$ single crystal was measured using a steady-state four-contact method on a custom-built small sample experiment mounted on a furnace which integrates a calibrated heat-pipe in series with the sample.

### Resistivity
Resistivity measurements were carried out using a SQUID PPMS-9 system (2–400 K, 0–9 T) and a home-made low temperature and high magnetic field transport measurement system (300 mK–300 K,

0–18 T). Temperature dependence of the in-plane resistivity $\rho_{ab}(T)$ of NbSe$_2$ and (NaOH)$_{0.5}$NbSe$_2$ samples were measured in a standard four-probe configuration with the applied current less than 2 mA. The out-of-plane resistivity $\rho_c(T)$ of NbSe$_2$ and (NaOH)$_{0.5}$NbSe$_2$ samples were measured using the Montgomery methods. Measurements in magnetic fields up to 18 T were conducted.

## Data availability
The data that support the findings of this study are available from the corresponding authors upon reasonable request.

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

## Acknowledgements

This work is financially supported by the Development Program of China (Grants No. 2018YFE0202600, X.L.C.); National Natural Science Foundation of China (Grants No. 52102338, R.J.S., 51772333, 52171229 S.F.J.); the National Key Research and the Key Research Program of Frontier Sciences, CAS, Grant No. QYZDJ-SSW-SLH013, S.F.J. and X.L.C. the Youth Innovation Promotion Association of CAS (Grant No. 2019005, 2021008 S.F.J. and Z.L.L.).

## Author contributions

R.J.S., S.F.J., and X.L.C conceived and designed the experiments. R.J.S., M.N.H., K.M., Y.X.M., C.C.Z., D.Z.M., and Y.Y.D. synthesized the crystals and performed basic physical and structural characterization. G.L. and X.J. performed ultra-low-temperature magnetic resistance measurements. J.D. and S.X.D. performed DFT calculations. Z.L.L performed the heat capacity measurement, X.W.W., Y.P., X.Q., R.G.Y., G.D.L., L.J.D., and H.Z.Z. performed the heat conductivity measurement, S.F.J., R.J.S., T.P.Y., and X.L.C. wrote the manuscript with contributions from all authors.

## Competing interests

The authors declare no competing interests.
