## [Peer Review File · Nature Communications]

High anisotropy in electric and thermal conductivity through the design of aerogel-like superlattice $(\text{NaOH})_{0.5}\text{NbSe}_2$REVIEWER COMMENTS

Reviewer #1 (Remarks to the Author):

This is a very interesting paper that clearly demonstrates the interlayer decoupling of bulk materials through the design of aerogel-like superlattice. The electric and vibration decoupling in $(\text{NaOH})_{0.5}\text{NbSe}_2$ is demonstrated through an unusual 1D insulating behavior, the absence of interlayer Raman modes, dominant local modes in heat capacity, significantly reduced interlayer coupling energy and ultralow interlayer thermal conductivity (0.28 W/mK at RT). The enhancement of CDW transition temperature (>110 K) and Pauling-breaking 2D superconductivity further suggested that bulk $(\text{NaOH})_{0.5}\text{NbSe}_2$ behaves similarly to an exfoliated NbSe_2 monolayer. The report of successful realization of electric & vibration decoupling in bulk material is significant, and the concept of introducing hollow, aerogel-like layers to decouple the bulk materials can be general. This study is new and will be of interest for a large community in condensed matter, it thus deserves publication as soon as possible.

Nevertheless, I have some remarks.

1. Giant anisotropy in electric and thermal conductivity is expected in a bulk material with interlayer decoupling. While the authors show a record low thermal conductivity of $(\text{NaOH})_{0.5}\text{NbSe}_2$ in the interlayer direction, which is about 5% of the value of conventional NbSe_2 , the thermal conductivity in the in-plane direction is missing.
2. Devarakonda, A. et al. grew a superlattice material $\text{Ba}_6\text{Nb}_{11}\text{S}_{28}$, and the material shows clean 2D superconductivity. (Science 2020, 370, 6513) The results are similar to the 2D superconductivity reported here in this paper, I would be interested to see the similarities and differences between them. Moreover, various mechanisms have been proposed for the superconductivity in monolayer NbSe_2 , the authors should give a more in-depth discussion on the Pauli-breaking superconductivity in $(\text{NaOH})_{0.5}\text{NbSe}_2$.

Here are some minor comments:

1. Considering XPS is a surface sensitive technique and the depths analyzed are limited to a few nanometers, did the XPS measurements in Fig 2(b) was an average result on a freshly cleaved crystal?
2. In Fig 2(c&d), which ones of the atoms are the Nb, Se, Na, and the O?
3. In Fig 3(a), the author exhibited the temperature dependent in-plane and out-plane resistivity measured for NbSe_2 and $(\text{NaOH})_{0.5}\text{NbSe}_2$ using log coordinates. The ordinary coordinates should also be shown to distinguish the superconducting transition temperature.
4. Comments on the Supplement, the color bars are missing in the EDS mapping present in Fig S2(b).

Reviewer #2 (Remarks to the Author):

The present manuscript describes the synthesis and characterization of a layered bulk material with alternating layers of NbSe_2 and a NaOH . The authors provide convincing evidence that the NbSe_2 layers are electronically and to a considerable extent also vibrationally decoupled from each other, leading to a strong anisotropy in the electrical conductivity. They also show data that supports the notion that the NbSe_2 layers in this bulk behavior exhibit similar electronic and superconducting properties as isolated NbSe_2 layers.

The results are interesting and potentially suitable for publication in Nature Communications.

There are, however, several issues that need considerable attention before this manuscript is at that level. In particular, the current title, abstract, and conclusions make claims that are not or incompletely supported by the facts presented.

1. The title claims "record high anisotropy in electrical and thermal conductivity".

* For the electrical conductivity the authors provide data for the in-plane and through-plane conductivity. The anisotropy is indeed very high. There is, however, no comparison with any other material or published data, which is clearly needed. Even if that is included claims of "a record" must be handled with extreme care.

* In the case of the thermal conductivity, the authors report the through-plane conductivity but no data for the in-plane conductivity. The anisotropy is thus unknown. Moreover the through-plane conductivity is actually not particularly low, certainly one cannot claim it to be "ultralow". There are various examples of even smaller conductivities in bulk materials, see, e.g., the classical paper by Cahill and co-workers on disordered, layered WSe₂ crystals, which reaches 50 mW/m/K at 300 K (and even smaller below). The comparison with literature is rudimentary and the anisotropy is likely to be smaller than published data.

* Line 154: The authors obtain a "fitted band gap of 0.5 eV". What is the physical significance of this value if any?

* Line 226: The authors refer to an inset in Figure 4d that does not seem to exist.

* Line 231: The referral of the lower through-plane conductivity to lattice mismatch in heterostructures is misleading. The low occupancy of the NaOH layer in the present case has pretty much the same effect of lowering the strength of the vdW interactions. Hence such a distinction is not helpful.

* Line 239: The authors refer to a "survey of layered vdW materials" in Figure 4d. The figure contains 3 more materials and only their through-plane conductivity. This hardly qualifies as a survey.

* The section on CDW (line 244ff, in particular line 258ff) is very heavy on technical details and difficult to follow. The subsequent paragraph on lines 266ff is even more strenuous to read. This part can and should be much improved, as there are important points embedded in this description.

* Figure 2: Please indicate which colors correspond to which species.

* The manuscript requires careful proofreading to improve the grammar, as the text is in parts difficult to parse.

Response Letter:

Response to Reviewer #1

This is a very interesting paper that clearly demonstrates the interlayer decoupling of bulk materials through the design of aerogel-like superlattice. The electric and vibration decoupling in $(\text{NaOH})_{0.5}\text{NbSe}_2$ is demonstrated through an unusual 1D insulating behavior, the absence of interlayer Raman modes, dominant local modes in heat capacity, significantly reduced interlayer coupling energy and ultralow interlayer thermal conductivity (0.28 W/mK at RT). The enhancement of CDW transition temperature (>110 K) and Pauling-breaking 2D superconductivity further suggested that bulk $(\text{NaOH})_{0.5}\text{NbSe}_2$ behaves similarly to an exfoliated NbSe_2 monolayer. The report of successful realization of electric & vibration decoupling in bulk material is significant, and the concept of introducing hollow, aerogel-like layers to decouple the bulk materials can be general.

This study is new and will be of interest for a large community in condensed matter, it thus deserves publication as soon as possible.

Reply: We thank the referee for the high evaluation and positive comments.

Nevertheless, I have some remarks.

1. Giant anisotropy in electric and thermal conductivity is expected in a bulk material with interlayer decoupling. While the authors show a record low thermal conductivity of $(\text{NaOH})_{0.5}\text{NbSe}_2$ in the interlayer direction, which is about 5% of the value of conventional NbSe_2 , the thermal conductivity in the in-plane direction is missing.

Response: We thank the referee for the comment. Indeed, the thermal conductivity in the in-plane direction is necessary to access the anisotropy value. In the revised manuscript, we have performed additional experiments to measure the in-plane thermal conductivity of $(\text{NaOH})_{0.5}\text{NbSe}_2$. Considering the beam-offset time domain thermos-reflectance (TDTR) method we initially used is optimized for measuring κ_{\perp} of a thin single crystal (Figure R1a), the in-plane thermal conductivity of $(\text{NaOH})_{0.5}\text{NbSe}_2$ is here obtained based on the steady-state method using a steady-state heat flow setup (inset of Figure R1b), so that the highest accuracy can be guaranteed for κ_{\parallel} . The in-plane thermal conductivity of $(\text{NaOH})_{0.5}\text{NbSe}_2$ was measured in the temperature range of 150 to 400 K, and the results are presented in Figure 4d and Figure R1b. At 300 K, the measured κ_{\parallel} is $98.01 \text{ W m}^{-1} \text{ K}^{-1}$, indicating that the thermal conductivity anisotropy $\kappa_{\parallel}/\kappa_{\perp}$ of $(\text{NaOH})_{0.5}\text{NbSe}_2$ reaches an exceptionally high value up to

350. This anisotropic value is surpassed only by misoriented r-MoS₂ thin films ($\kappa_{\parallel}/\kappa_{\perp} \approx 880$, *Nature* **2021**, 597, 660), and exceeds that of pyrolytic graphite (PG), which is widely acknowledged as one of the most anisotropic bulk thermal conductors ($\kappa_{\parallel}/\kappa_{\perp} \approx 340$, *J. Phys. Chem. 1*, **1972**, 279–421).

Action: We have added the in-plane thermal conductivity in Fig 4d and the thermal conductivity anisotropy in Fig 4e. Two authors that contribute to the steady-state heat flow experiment are added to the author list.

Fig. R1 (a) TDTR data for the out-of-plane thermal conductivity κ_{\perp} of $(\text{NaOH})_{0.5}\text{NbSe}_2$. (b) The in-plane thermal conductivity κ_{\parallel} of $(\text{NaOH})_{0.5}\text{NbSe}_2$ measured by steady-state heat flow method.

2. Devarakonda, A. et al. grew a superlattice material $\text{Ba}_6\text{Nb}_{11}\text{S}_{28}$, and the material shows clean 2D superconductivity. (*Science* 2020, 370, 6513) The results are similar to the 2D superconductivity reported here in this paper, I would be interested to see the similarities and differences between them. Moreover, various mechanisms have been proposed for the superconductivity in monolayer NbSe_2 , the authors should give a more in-depth discussion on the Pauli-breaking superconductivity in $(\text{NaOH})_{0.5}\text{NbSe}_2$.

Response: We thank the reviewer for bringing up this relevant study. In $\text{Ba}_6\text{Nb}_{11}\text{S}_{28}$, the introduction of 1 nm thick, stoichiometric Ba_3NbS_5 layers electrically dissociates the superconducting NbS_2 layers from each other, resulting in clean Pauli-breaking 2D superconductivity. In contrast, $(\text{NaOH})_{0.5}\text{NbSe}_2$ consists of 2 nm thick NaOH block layers, which exhibit a flurry and aerogel-like structure, leading to significantly higher off-plane resistivity and anisotropy. For instance, in terms of anisotropy in both normal state resistivity (β) and superconductivity upper critical fields (γ), the values in $(\text{NaOH})_{0.5}\text{NbSe}_2$ ($\beta_{\text{max}} > 2,000,000$, $\gamma = 32.5$) are superior to those in the 'layer-in-cake' 2D

superconductors $\text{Ba}_6\text{Nb}_{11}\text{S}_{28}$ ($\beta_{\text{max}} \approx 2,000$, $\gamma = 26.34$) and its sister compound $\text{Ba}_6\text{Nb}_{11}\text{Se}_{28}$ ($\beta_{\text{max}} \approx 20.0$, $\gamma = 10.3$). The Pauli violation ratio (PVR) in $(\text{NaOH})_{0.5}\text{NbSe}_2$ ($P = 4$) also approaches that of monolayer NbSe_2 (PVR = 6), and much higher than that in $\text{Ba}_6\text{Nb}_{11}\text{S}_{28}$ (PVR = 1.5). Moreover, the off-stoichiometric and aerogel-like NaOH layers in $(\text{NaOH})_{0.5}\text{NbSe}_2$ enable vibrational decoupling and enhance the charge density wave (CDW) temperature, which mimics the behavior of monolayer NbSe_2 but differs significantly from the findings in stoichiometric $\text{Ba}_6\text{Nb}_{11}\text{S}_{28}$ and $\text{Ba}_6\text{Nb}_{11}\text{Se}_{28}$.

The strong Pauli-breaking Ising superconductivity observed in monolayer NbSe_2 has garnered significant attention. More recently, the Pauli-breaking Fulde–Ferrell–Larkin–Ovchinnikov (FFLO) state and orbital-FFLO have also been discovered in $\text{Ba}_6\text{Nb}_{11}\text{S}_{28}$ (*Science* 2020, 370, 6513) and multilayer NbSe_2 (*Nature* 2023, 619, 46). In comparison to these $\text{NbS}_2/\text{NbSe}_2$ -based 2D superconductors, the behavior of $(\text{NaOH})_{0.5}\text{NbSe}_2$ closely resembles that of monolayer NbSe_2 . For instance, the Pauli violation ratio in $(\text{NaOH})_{0.5}\text{NbSe}_2$ is second only to monolayer NbSe_2 , and approximately 2.5 times higher than the FFLO systems $\text{Ba}_6\text{Nb}_{11}\text{S}_{28}$ and multilayer 2H- NbSe_2 . Additionally, in clean FFLO systems (e.g., $\text{Ba}_6\text{Nb}_{11}\text{S}_{28}$, multilayer NbSe_2), the $\mu_0 H c_2$ transition typically exhibits an anomalous sharpening as the Pauli limit is approached within a critical region $|\theta - 90^\circ| < \delta\theta$. However, this anomalous enhancement of $\mu_0 H c_2(\theta)$ after crossing the Pauli paramagnetic limit is not observed in $(\text{NaOH})_{0.5}\text{NbSe}_2$ and monolayer NbSe_2 . Lastly, the anisotropy of the superconducting state in $(\text{NaOH})_{0.5}\text{NbSe}_2$ ($\gamma = 32.5$) is nearly identical to that in monolayer NbSe_2 ($\gamma = 33$). These findings suggest that $(\text{NaOH})_{0.5}\text{NbSe}_2$ behaves as an Ising superconductor without FFLO and orbital-FFLO states, similar to monolayer NbSe_2 .

Action: In the revised manuscript, we have included a comparison of the anisotropy and 2D superconductivity between $(\text{NaOH})_{0.5}\text{NbSe}_2$ and the findings in $\text{Ba}_6\text{Nb}_{11}\text{S}_{28}$ by Devarakonda et al. Additionally, we have expanded the discussion on the mechanisms of superconductivity in monolayer NbSe_2 , providing a more comprehensive explanation of the observed Pauli-breaking superconductivity in $(\text{NaOH})_{0.5}\text{NbSe}_2$ (page 14).

Here are some minor comments:

1. Considering XPS is a surface sensitive technique and the depths analyzed are limited to a few nanometers, did the XPS measurements in Fig 2(b) was an average result on a freshly cleaved crystal?

Response: The XPS measurement is a surface sensitive method, and to ensure accuracy, the result in Fig 2(b) was indeed carried out on a freshly cleaved

single crystal.

Action: We have clarified this point in the revised manuscript (Section 2.2).

2. In Fig 2(c&d), which ones of the atoms are the Nb, Se, Na, and the O?

Response: We apologize for the confusion. In Fig 2(c&d), the atoms are labeled as follows: Nb (blue), Se (yellow), Na (green), and O (red).

Action: We have added a corresponding legend to clarify this in Fig. 2.

3. In Fig 3(a), the author exhibited the temperature dependent in-plane and out-plane resistivity measured for NbSe_2 and $(\text{NaOH})_{0.5}\text{NbSe}_2$ using log coordinates. The ordinary coordinates should also be shown to distinguish the superconducting transition temperature.

Response: We appreciate the reviewer's suggestion.

Action: We have included additional figures showing the temperature-dependent resistivity using ordinary coordinates to show the superconductivity (Figure S4b).

4. Comments on the Supplement, the color bars are missing in the EDS mapping present in Fig S2(b).

Response: We thank the referee for pointing out this.

Action: As suggested, we have incorporated color bars into Figure S2(b).

Reviewer #2 (Remarks to the Author):

The present manuscript describes the synthesis and characterization of a layered bulk material with alternating layers of NbSe₂ and a NaOH. The authors provide convincing evidence that the NbSe₂ layers are electronically and to a considerable extent also vibrationally decoupled from each other, leading to a strong anisotropy in the electrical conductivity. They also show data that supports the notion that the NbSe₂ layers in this bulk behavior exhibit similar electronic and superconducting properties as isolated NbSe₂ layers.

The results are interesting and potentially suitable for publication in Nature Communications. There are, however, several issues that need considerable attention before this manuscript is at that level. In particular, the current title, abstract, and conclusions make claims that are not or incompletely supported by the facts presented.

We thank the referee for the positive comments and appreciation of our work.

1. The title claims "record high anisotropy in electrical and thermal conductivity".

* For the electrical conductivity the authors provide data for the in-plane and through-plane conductivity. The anisotropy is indeed very high. There is, however, no comparison with any other material or published data, which is clearly needed. Even if that is included claims of "a record" must be handled with extreme care.

Response: We appreciate the reviewer's valuable comment. In the revised manuscript, we have included a comparison of the electrical conductivity anisotropy in (NaOH)_{0.5}NbSe₂ with well-known metallic materials that exhibit notable anisotropy, such as Graphite, TMDs, molecular intercalated TMDs, the misfit compounds with TMDs layers, and electrically decoupled Ba₆Nb₁₁S₂₈. We have also provided a more cautious interpretation of our results. As depicted in Figure R2, the maximum resistivity anisotropy in (NaOH)_{0.5}NbSe₂ reached an exceptionally high value of 2x10⁶, surpassing the highest values reported in the aforementioned anisotropic materials by two or three magnitudes. To address the reviewer's concerns, we have removed the claims of 'a record high' from the title.

Action: As shown below (Fig.R2), the resistance anisotropy of (NaOH)_{0.5}NbSe₂ with its through-plane conductivity determined by Montgomery method is compared with various materials in Fig. S5. Moreover, we have removed the "a record" from the title, and changed it to "High anisotropy in electric and thermal conductivity through the design of aerogel-like superlattice (NaOH)_{0.5}NbSe₂".

Fig. R2 Resistivity anisotropy β of metallic TMDs, their molecular intercalants EDA-TaS₂/NbSe₂, Pyr-TaS₂, inorganic intercalants (SnSe)_{1.16}NbSe₂, (BiSe)_{1.16}NbSe₂, Ba₆Nb₁₁S/Se₂₈, graphite and (NaOH)_{0.5}NbSe₂. (Reference: *Science* 2020, 370, 6513; *Phys. Rev. Materials* 6, 044806, *The Journal of Chemical Physics* 62, 4411 (1975); *Proc. R. Soc. Lond. A* 1972 327, 289-303)

* In the case of the thermal conductivity, the authors report the through-plane conductivity but no data for the in-plane conductivity. The anisotropy is thus unknown. Moreover the through-plane conductivity is actually not particularly low, certainly one cannot claim it to be "ultralow". There are various examples of even smaller conductivities in bulk materials, see, e.g., the classical paper by Cahill and co-workers on disordered, layered WSe₂ crystals, which reaches 50 mW/m/K at 300 K (and even smaller below). The comparison with literature is rudimentary and the anisotropy is likely to be smaller than published data.

Reply: We agree with the referee that measuring the thermal conductivity in the in-plane direction is crucial to determine the anisotropy value. We have conducted additional experiments to measure the in-plane thermal conductivity. To ensure the highest accuracy, the off-plane thermal conductivity (κ_{\perp}) of (NaOH)_{0.5}NbSe₂ single crystal is measured using the beam-offset time domain thermos-reflectance (TDTR) method, while a steady-state heat flow setup (Fig. R3b) based on steady-state method is employed to obtain the in-plane thermal conductivity (κ_{\parallel}) between 150-350 K (Figure R3a). At 300 K, the measured κ_{\parallel} is 98.1 W/mK, and the thermal conductivity anisotropy Δ of (NaOH)_{0.5}NbSe₂ reaches an exceedingly high value up to 350.4.

Fig. R3 (a) Raw data of the static-method thermal conductivity measurement of $(\text{NaOH})_{0.5}\text{NbSe}_2$. (b) The in-plane thermal conductivity data between 150-350 K. (c) TDTR data for the out-of-plane thermal conductivity κ_{\perp} at 300 K.

We also thank the reviewer for bringing up the relevant results of Cahill and co-workers. As shown in Figure R4a, we find this ultralow κ_{\perp} of 50 mW/m/K is realized on thin films of disordered WSe_2 , while in bulk WSe_2 single crystals the κ_{\perp} value is increased significantly to 1.5 ~3 W/m/K. We have followed the suggestion of the referee to give more comprehensive information on the materials with ultralow thermal conductivity and high anisotropy. In bulk crystalline materials, the progress in pushing thermal conductivity to its lowest limit is realized in Tl_3VSe_4 (0.3 W/m/K, powder sample, *Science* 2018, 360, 1455), $\text{BiO}_2\text{Cl}_2\text{Se}$ (0.1 W/m/K, powder sample, *Science* 2021, 373, 1017) and recently in Ag_8SnSe_6 that with an “ultralow” conductivity of 0.5 W/m/K as claimed by the authors (Qingyong Ren et al. *Nat. Mater.* 2023). Therefore, the through-plane conductivity of 0.28 W/m/K in $(\text{NaOH})_{0.5}\text{NbSe}_2$ is among the lowest values that achieved in bulk inorganic materials. We have added the relevant references to the manuscript, and to address the concern of the referee, we have removed the phrase “ultralow” from the abstract and the main text.

In Figure R4b, we have also summarized the thermal conductivity anisotropy Δ of $(\text{NaOH})_{0.5}\text{NbSe}_2$ and other layered materials reported in a relevant literature (*Nature* 2021, 597, 660). It is found that the thermal conductive anisotropy of 350 in $(\text{NaOH})_{0.5}\text{NbSe}_2$ is currently the highest value that has been achieved in

bulk crystalline materials. The value exceeds that of pyrolytic graphite (PG), which is widely acknowledged as one of the most anisotropic bulk thermal conductors ($\Delta \approx 340$). Taking thin films into consideration, the value is second only to that reported by Shi En Kim et al. in an extremely anisotropic thermal conductor, i.e., MoS₂ films stacked by random monolayers ($\Delta \approx 880$). In addition, the results of disordered, WSe₂ thin films reported by Cahill and co-workers is also included in the diagram. We find the reported anisotropy value for misoriented WSe₂ films is moderately large, $\Delta \approx 30$, which is one-order of magnitude smaller than that of (NaOH)_{0.5}NbSe₂.

Action: We have added the in-plane thermal conductivity data in Fig 4d and the thermal conductivity anisotropy data in Fig 4e. According to the suggestions of the referee, we also removed the phrase “ultralow off-plane thermal conductivity”, and replaced it by “low off-plane thermal conductivity” in both the abstract and main text. The relevant results of Cahill and co-workers and Shi En Kim, etc. are also cited.

Fig. R4 (a) Summary of measured thermal conductivities of WSe₂ films and bulk crystal (taken from Cahill et al. *Science* 2007, 315, 5810, 351-353). (b) Comparison of anisotropy $\Delta = \kappa_{\parallel} / \kappa_{\perp}$ (y axis), κ_{\perp} (x axis), and κ_{\parallel} (diagonal dashed lines) measured for highly anisotropic thermal conductors (taken from *Nature* 2021, 597, 660) and (NaOH)_{0.5}NbSe₂.

* Line 154: The authors obtain a "fitted band gap of 0.5 eV". What is the physical significance of this value if any?

Response: That is a very interesting question. Typically, in conventional materials, the crystal is either a metal or a semiconductor. However, the resistivity behavior in $(\text{NaOH})_{0.5}\text{NbSe}_2$ is axially dependent, making it more complex. The bulk superlattice is formed by alternately stacking metallic NbSe_2 layers with thick semiconducting NaOH layers. As a result, the equivalent circuit of the material along the c-axis consists of a series connection of metals and semiconductors. The semiconducting behavior observed in the $\rho_c(T)$ curve is likely due to the 2 nm thick NaOH block layers, suggesting that the NbSe_2 layers are indeed electrically decoupled. Considering the fitted band gap value might be misunderstood to the actual band gaps in a conventional semiconductor, we have opted to remove this value.

Action: We have deleted the misleading bandgap value in the revised version.

* Line 226: The authors refer to an inset in Figure 4d that does not seem to exist.

Response & Action: We apologize for the confusion. This typo has been corrected in Figure S7.

* Line 231: The referral of the lower through-plane conductivity to lattice mismatch in heterostructures is misleading. The low occupancy of the NaOH layer in the present case has pretty much the same effect of lowering the strength of the vdW interactions. Hence such a distinction is not helpful.

Response: We agree with the reviewer that such a distinction is not helpful here. The referral of lattice mismatch in heterostructures has been removed.

Action: We have removed the relevant statement in the main text.

* Line 239: The authors refer to a "survey of layered vdW materials" in Figure 4d. The figure contains 3 more materials and only their through-plane conductivity. This hardly qualifies as a survey.

Response: Thank you for pointing this out. In the new Figure 4f, we have summarized the results of nine typical layer materials, providing more comprehensive information for the reader.

Action: We have added 6 more data in Figure 4f.

* The section on CDW (line 244ff, in particular line 258ff) is very heavy on

technical details and difficult to follow. The subsequent paragraph on lines 266ff is even more strenuous to read. This part can and should be much improved, as there are important points embedded in this description.

Response: We appreciate the reviewer's suggestions.

Action: In the revised manuscript, we have provided more background information and simplified the section on CDW to enhance its readability, while still conveying our key findings.

* Figure 2: Please indicate which colors correspond to which species.

Response: We appreciate the referee's careful examination of our manuscript.

Action: In the revised version, we have added labels to identify the atomic species in Figures 2c and 2d.

* The manuscript requires careful proofreading to improve the grammar, as the text is in parts difficult to parse.

Response: In the revised manuscript, we have carefully proofread the text to improve the grammar and readability, thank you!

REVIEWERS' COMMENTS

Reviewer #1 (Remarks to the Author):

The authors have sufficiently addressed my questions. I enthusiastically support the publication of the manuscript.

Reviewer #2 (Remarks to the Author):

The authors have adequately addressed the comments raised in the first round of review and extended their manuscript accordingly. I consider the manuscript suitable for acceptance now.

Response Letter:

Response to Reviewer #1

The authors have sufficiently addressed my questions. I enthusiastically support the publication of the manuscript.

Reply: We thank Reviewer #1 for recommending the publication of our work.

Response to Reviewer #2

The authors have adequately addressed the comments raised in the first round of review and extended their manuscript accordingly. I consider the manuscript suitable for acceptance now.

Reply: We thank Reviewer #2 for recommending the publication of our work.